# Constructing a Double Alternant “Rigid-Flexible” Structure for Simultaneously Strengthening and Toughening the Interface of Carbon Fiber/Epoxy Composites

**DOI:** 10.3390/nano12173056

**Published:** 2022-09-02

**Authors:** Susu Zhang, Ping Han, Lina Yang, Shaokai Hu, Jianfa Wang, Zheng Gu

**Affiliations:** 1College of Chemistry and Chemical Engineering, Qingdao University, Qingdao 266071, China; 2College of Physics, Qingdao University, Qingdao 266071, China; 3Weihai Innovation Institute, Qingdao University, Weihai 264200, China; 4Weifang Key Laboratory of Environmentally Friendly Macromolecular Flame Retardant Materials, Weifang 262715, China; 5Shandong Engineering Laboratory of Environmentally Friendly Macromolecular Flame Retardant Materials, Weifang 262715, China

**Keywords:** carbon fibers, epoxy composites, double alternant “rigid-flexible”, strength, toughness

## Abstract

An optimized “rigid-flexible” structure with multistage gradient modulus was constructed on carbon fiber (CF) surface via chemical grafting using “flexible” polyethyleneimine (PEI) and “rigid” polydopamine (PDA) between “rigid” CF and “flexible” epoxy (EP) to elaborate a double alternant “rigid-flexible” structure for simultaneously strengthening and toughening CF/EP composites. PDA and PEI polymers can greatly enhance the roughness and wettability of CF surfaces, further strengthening the mechanical interlocking and chemical interactions between CFs and epoxy. Besides, the “rigid-flexible” structure endows the interface with a gradient transition modulus, which could uniformly transfer internal stress and effectively avoid the stress concentration. Moreover, the double alternant “rigid-flexible” could buffer the external loading, induce more micro cracks and propagation paths and, thereby, consume more energy during the destruction of the composite. The interfacial shear strength, interlaminar shear strength, impact strength increased by 80.2%, 23.5% and 167.2%, and the fracture toughness improved by 227.2%, compared with those of the unmodified CF composite, respectively. This creative strategy and design afford a promising guidance for the preparation and production of advanced CF/EP structural materials with high strength and toughness.

## 1. Introduction

Carbon fiber (CF) reinforced polymer composites (CFRPs) have been widely applied in aerospace, defense, automotive and electronic equipment, due to their light weight, high strength and erosion resistance [1,2,3,4]. Despite its outstanding advantages, the potential of CFRP has not been exploited fully yet due to its nonpolarity and chemical inert surface, as well as the great modulus difference with the matrix [5]. Usually, the interface between reinforcement and matrix plays a decisive role in the comprehensive properties of the composites. Strength and toughness are contradictory performances for structural materials, and an increase in one always leads to a decrease in another [6]. Strong interface adhesion is beneficial to transfer stress, thereby improving the strength of composites [7], while weak interface bonding is conductive to an increase in the interface toughness by means of plastic deformation of the interface and the energy consumption during the composites fracture [8]. Therefore, how to design and construct an optimized interphase for simultaneously improving the strength and toughness is imperative for structural materials.

To date, various CF surface modification methods have been established, such as bio-spired, chemical grafting, coating and electrophoretic deposition [9,10,11,12]. Among them, chemical grafting is considered as an effective method because of its simple operation, low-energy nature and cost-effectiveness.

The “rigid-flexible” architectures inspired by the nacre have drawn a lot of attentions because it can endow composites with prominent mechanical strength and toughness [13]. Inorganic nanoparticles are usually selected as “rigid” components to incorporate in the interface for improving mechanical strength, such as graphene, nanoclay, carbon nanotube (CNT) and TiO_2_ [14,15,16,17]. Nevertheless, in most cases, nanoparticles need to be pretreated in order to introduce active sites to increase the interaction with carbon fiber and matrix resin. On the other hand, most nanoparticles tend to aggregate due to the incompatibility with the matrix, consequently, leading to stress concentration and degeneration of the composite performance. Polymers introduced in the interface as “flexible” components have been proved to be an effective method to improve the interfacial toughness via inducing viscoplastic deformation, microcracks and intermolecular entanglement [18], while this process always accompanied with the reduction in mechanical strength of the composites.

The mussel-inspired polydopamine (PDA) owns abundant active groups and rigid structure. Besides, due to the excellent adhesive property, PDA is often used as the secondary reaction platform [19,20]. Jin et al. [21] chose Polydopamine (PDA) as a functional platform to integrate rigid Ni(OH)_2_ nanosheets and amine-rich polymer onto CFs for constructing a sandwich-like “rigid-soft” interface. The interlaminar shear strength (ILSS) of the modified composites improved up to 43.38%, compared with the unmodified composites. Wu et al. [22] constructed a nacre-like interphase by alternatively depositing “rigid” polydopamine (PDA) and “flexible” polyether amine on carbon fiber surface via the layer-by-layer (LbL) approach. The interfacial strength and toughness increased by 39.2% and 99.8% compared with the unmodified composites, respectively.

In this work, we put forward a prospective double alternant “rigid-flexible” structure: inserting “rigid-flexible” interfacial layer between “rigid” CF and “flexible” epoxy via simple chemical grafting technique. PDA was used as rigid component rather than a secondary reaction platform. It can enhance the strength of the interface due to abundant aromatic structures in its molecular chain, on the one hand; on the other hand, its implementation can be effectively compatible with the matrix because of its polymer characteristic, thus avoiding the aggregation results of inorganic nanomaterials. Besides, abundant active groups in PDA molecules can react with amino-containing polymers to enhance the interface adhesion. Polyethyleneimine (PEI) is a 3-dimensional hyperbranched polymer containing a variety of polar amine groups and longer molecular chains. It has been proved that modifying CF with PEI could significantly improve the interface properties and mechanical properties of the composites [23]. It can react with most oxygen-containing functional groups and reactive groups in PDA, thereby realizing a “soft” bridge between CFs and PDA. Besides, the longer molecular chains could form a network structure and complex intermolecular chain entanglement with PDA and epoxy. Most importantly, the double alternant “rigid-flexible” structure could effectively buffer and relieve the external loading, induce plastic deformation and more microcracks, thereby consuming more energy during the destruction process of the composites under loading, eventually improve the strength and toughness of the composites.

The morphologies of the CF surface and composite fracture surface were characterized by SEM, TEM and AFM. The chemical composition and structure were validated by Raman, XPS spectra and EDS. The wettability of fibers with epoxy matrix was estimated by dynamic contact angle (DCA) analysis. The strength and toughness of the composite were represented by the interfacial shear strength (IFSS), interlaminar shear strength (ILSS), impact strength and the interfacial toughness. Besides, the mechanism of the performance improvement was deeply discussed based on the fracture interface of the micro-debonding test.

## 2. Materials and Methods

### 2.1. Materials

Polyacrylonitrile-based commercial CFs with average diameter of 7 μm and density of 1.76 g·cm^−3^ were provided by the Sino Jilin Carbon Co., Jilin, China. Dopamine hydrochloride (DA) and polyethyleneimine (PEI, molecular weight 400) were produced by Shanghai Aladdin Reagent Co., Ltd. (Shanghai, China). The purity of dopamine was 98% and its relative molecular mass was 189.64. Tris-(hydroxymethyl) aminomethane hydrochloride (Tris-HCl, 99% purity) was marketed from Shanghai Shanpu Chemical Co., Ltd. (Shanghai, China). N,N′-dicycclohexyl carbonimide (DCC) as dehydration condensation agent was provided by Shanghai Aladdin Reagents. The epoxy resin and curing agent used in this work were E-51 (molecular weight 350~400, epoxide number 0.51) and 4,4′-Methylenebis(2-ethylaniline) (H-256), which were bought from Wuxi Resin Factory and Hubei Johnson Technology Co., Ltd., (Wuhan, China). respectively. Potassium persulfate (K_2_S_2_O_8_), silver nitrate (AgNO_3_), acetone, ethanol and N, N-Dimethylformamide (DMF) were purchased from the Sinopharm Chemical Reagent Co., Ltd. (Shanghai, China).

### 2.2. Preparation of Functionalized Fibers

#### 2.2.1. Preparation of the CF-PEI

A certain amount of carbon fibers were washed in the acetone at 333 K for 48 h, which was named unmodified CF. Subsequently, the unmodified CF was treated in the oxidant system of AgNO_3_ (0.01 mol/L)/K_2_S_2_O_8_ (0.1 mol/L) solution at 343 K for 1 h to introduce oxygen-containing polar groups onto CF surface and control the strength loss of the oxidized carbon fibers at the same time. PEI was dissolved in the N, N-Dimethylformamide (DMF) solution in a volume ratio of 1:10 under ultrasound, then CF-COOH was placed into above mixed solution and oil bathed for 24 h with DCC as a dehydration agent [23]. The treated substrate was then refluxed with alcohol for 2 h to remove residual solvent and unreacted PEI on the CF surface. Finally, the fibers were put into a dry oven at 353 K for 12 h.

#### 2.2.2. Preparation of CF-PEI-PDA

0.01 mol·L^−1^ Tris buffer was configured with deionized water until the PH of the solution reached 8.5. Then, the CF-PEI was immersed into a buffer solution of DA (2 mg·mL^−1^) at 298 K for 24 h with stirring, followed by being washed with deionized water at 298 K and dried for 12 h at 353 K. The resulted sample was named CF-PEI-PDA.

#### 2.2.3. Preparation of Fiber/Epoxy Composite

To produce the composite, firstly, the epoxy (M_E-51_ = 10 g) and curing agent (M_H-256_ = 3.2 g) were mixed to prepare resin matrix. Then different carbon fibers (20 cm × 25 laps) were infiltrated entirely in the matrix under vacuum. After that, the prepregs were put into the preheated mold and maintained at 363 K for 2 h, 393 K for 2 h and 423 K for 3 h, accompanied by the load pressures of 5, 10 and 10 MPa, respectively. After cooling to room temperature, the CF/EP composite was obtained. The whole process and possible reaction mechanisms were presented in Figure 1.

### 2.3. Characterization

The surface morphology of CFs and fractured topography of CF/EP composites were estimated by scanning electron microscopy (SEM, SU8010, KS, USA) with an accelerating voltage of 3 kV, transmission electron microscopy (TEM, JEM-2100F, Tokyo, Japan) with an operation voltage of 200 kV and atomic force microscopy (AFM, NT-MDT Co., Moscow, Russia) at a scanning speed of 1 Hz. Besides, AFM can also be used to measure the surface roughness of CFs and the modulus changes of the composites. The surface structure and chemical constitution of fibers were evaluated by Raman spectroscopy (RM2000, Renishaw, UK) with a laser excitation of 532 nm, X-ray photoelectron spectroscopy (XPS, ESCALAB220i-XL, Britain) with a laser excitation of 532 nm and Fourier transform infrared spectroscopy (FTIR, NEXUS, Perkin Elmer, Hopkinton, MA, USA) with a laser excitation of 532 nm. Three samples were tested per technique.

The TEM specimens were prepared as follows: Firstly, chopped fibers monofilaments were ultrasonic in ethanol solution for 15 min to acquire a uniform dispersion. Secondly, the dispersion was dripped onto a microgrid copper mesh and dried with an infrared lamp, and this progress was repeated three times. Finally, the copper mesh together with fibers was dried and ready for testing.

The wettability of the fibers was measured by a dynamic contact angle meter (DCAT21, Germany) with deionized water and diiodomethane as test liquids with 1200 g/m^3^ at 20 °C. The errors were calculated from the three samples tested. The CF surface energy is calculated according to Equations (1) and (2).
(1)γ1(1+cosθ)=2(γ1pγfp)12+2(γ1dγfd)12
(2)$$\gamma_{f}^{}=\gamma_{f}^{p}+\gamma_{f}^{d}$$
where, $\gamma_{1}$, $\gamma_{1}^{d}$ and $\gamma_{1}^{p}$ represents the surface tension, dispersion and polar component of the test liquids (deionized water and diiodomethane), respectively.

The interfacial shear strength (IFSS) was used to quantify the interfacial property between CFs and epoxy, which was tested by the monofilament debonding test equipment (FA620, Japan). The interlaminar shear strength (ILSS) and flexural tests were measured by a short beam shear test (GT-7000-A2X, China) according to ASTM D2344 and at a cross-head speed of 1 mm·min^−1^. The final IFSS and ILSS values all were the average of at least 25 valid data of three samples. The IFSS and ILSS values can be calculated according to Equations (3) and (4).
(3)$$\mathrm{IFSS}=\frac{F_{max}}{\pi d_{f}L_{e}}$$
where, $F_{max}$, $d_{f}$ and $L_{e}$ represent the maximal load (N), fiber diameter (mm) and length (mm) embedded in the epoxy droplet, respectively. The final IFSS value is the average of at least 25 valid data.
(4)$$ILSS=0.75\times \frac{F_{max}}{b\times h}$$
where $F_{max}$ is the breaking load (N), b and h are the specimen width and thickness (mm), respectively. The final ILSS value is the average of at least 10 valid data.

The toughness of composites was calculated according to the report of Scheer and Nairn:(5)$$F=\pi r_{1}^{2}\sqrt{\frac{2G_{ic}}{r_{1}G_{33s}}}$$
(6)$$G_{33s}=\frac{1}{2}(\frac{1}{E_{1}}+\frac{V_{1}}{V_{2}E_{2}})$$
(7)$$V_{1}=1.5{\left(\frac{r_{1}}{r_{2}}\right)}^{2}$$
where $G_{ic}$ is the fracture toughness, $E_{1}$ and $E_{2}$ are the Young’s moduli of fibers and matrix, $V_{1}$ and $V_{2}$ are the volume fractions of fiber and resin droplet ($V_{2}=1-V_{1}$), $r_{1}$ and $r_{2}$ are the radiuses of fiber and resin droplet, respectively.

The impact strength of composites was obtained by a drop weight impact test (9250 HV, Instron, MA, USA) with an impact velocity of 2 m·s^−1^ followed the ISO 179. The sizes of each specimen were 55 mm × 6 mm × 3 mm, and the average values of five samples were calculated.

The dynamic mechanical thermal properties were measured with a dynamic mechanical thermal analyzer (DMTA 8000, Hopkinton, MA USA) in a three-point bending mode during 50~200 °C with a heating rate of 5.0 °C·min^−^^1^ and load frequency of 1.0 Hz. The final results and error were obtained from five samples.

The interfacial microstructures between CFs and the matrix were estimated by the EDS techniques (JEOL JSM-5900LV) equipped on SEM basing on the difference of carbon element content between CFs and epoxy.

## 3. Results and Discussion

### 3.1. Surface Morphology of Carbon Fibers

The surface morphology of different fibers was shown in Figure 2. For unmodified CF, the surface roughness (Ra) was 79.9 nm, and abundant shallow grooves distributed parallel along the axial direction of CFs (Figure 2a,d,g). After grafting PEI, a thin layer of polymer uniformly covered the fiber surface, filling most of the grooves and resulting in a smoother surface with a lower Ra of 66.6 nm (Figure 2b,e,h). For CF-PEI-PDA (Figure 2c,f,i), plenty of PDA particles were scattered uniformly on the CF surface, forming a rougher surface with a larger Ra of 114.0 nm. The PDA particles can greatly increase the specific surface area of fibers and friction points, thereby improving the mechanical interlocking interaction between CF and epoxy matrix. Besides, the PEI and PDA constitute a “rigid-flexible” structure with a thickness of 326 nm, which can embed in the matrix as the intermediate interface layer.

### 3.2. Surface Structure and Chemical Composition of Carbon Fibers

Raman spectra are an effective means for analyzing the degree of order and graphitization for carbon materials. The Raman spectra results are presented in Figure 3a and Table 1. The Raman spectra of fibers shows two obvious peaks at 1330~1350 cm^−1^ and 1580~1600 cm^−1^, which are attributed to the defects together with disorder carbonaceous structure (D band) and ordered graphite structure (G band), respectively. The relative strength ratio of I_D_/I_G_ (R) is usually used to judge the defects degree of the carbonaceous structure. The D band and G band are prone to overlap, so a new peak (A band) is usually fitted in the range of 1500~1550 cm^−1^, which is believed to be attributed to the presence of amorphous carbon, heteroatom or some active functional groups. The relative strength ratio of I_A_/I_G_ indicates the extent of amorphous carbonaceous structure and the content of hetero atoms-containing functional groups on CF surface.

Comparing CF-COOH with unmodified CF, the R and I_A_/I_G_ values increased from 1.65 to 1.76 and 0.21 to 0.24, respectively, indicating that the degree of graphitization was reduced and the disorder level was increased. Meanwhile, amount of active oxygen-containing groups were introduced on the CF surface during the oxidizing process, which could provide more active points for the next grafting reaction. For CF-PEI, the I_D_/I_G_ grows to 1.82, due to the increased fiber surface disorder via grafting hyperbranched PEI, and the I_A_/I_G_ increased to 0.25 because of the introduction of more active nitrogen-containing groups. It is worth noting that the CF-PEI-PDA has the maximal values of I_D_/I_G_ (1.93) and I_A_/I_G_ (0.33), implying the most disordered surface structure and the most active functional groups on fibers surface. A number of hydrogen bonds, covalent bonds and π–π bonds between PEI and PDA make them constitute a stable “rigid-flexible” structure, which can further be assembled between CF and epoxy via mechanical interlocking and chemical bonding interaction to construct a double alternant “rigid-flexible” structure.

The chemical composition of different fibers was analyzed by FTIR (Figure 3b). Unmodified CF displays four typical peaks: -OH stretching vibration at 3440 cm^−1^, C–H stretching vibration of methyl and methylene at 2800~2980 cm^−1^, C=C stretching vibration at 1634 cm^−1^ and C–C stretching vibration at 1080 cm^−1^ [18]. After oxidation, the intensity of the peak at 1727 cm^−1^ became higher, which was attributed to the increase in C=O in carboxyl groups by oxidation. For CF-PEI, two new peaks appear at approximate 3230 cm^−1^ and 1385 cm^−1^, which belong the stretching resonance of N-H and the shearing vibration of C=N coming from the Schiff base reaction [24]. After PDA modification, more characteristic function groups of PDA emerged, including the stretching of C-H in the benzene ring at 3018 cm^−1^, the stretching vibration of indole ring at 1560 cm^−1^ and shearing vibration of aromatic C=N at 1390 cm^−1^ [25,26], suggesting the presence of PDA and a Schiff base reaction occurring on the fiber surface. The intensity of the peak at 2800–2980 cm^−1^ became higher, indicating the presence of a Michael reaction [15].

XPS was utilized to analyze the element content and chemical bonding states of CF surfaces and the results are shown in Table 2 and Figure 4. For unmodified CF, the elemental proportions of C, N and O are 93.71%, 1.80% and 4.49%, respectively. After grafting PEI, the N content grows to 9.10%, revealing the successful grafting of PEI. For CF-PEI-PDA, the N content increases substantially to 10.37%; at the same time, the O content increases to 21.79%, confirming the existence of N-rich and O-rich PDA layer. In other words, more of the active functional groups are grafted onto the carbon fiber surface by grafting PDA [27].

As shown in Figure 4a, the wide-scan spectrum for C, N and O elements were centralized at about 286.1, 400.1 and 531.1 eV, respectively. For unmodified CF (Figure 4b), the C1s spectra could be fitted into three peaks at 284.6, 285.6 and 286.3 eV, which are attributed to peak C1s (1), C1s (2) and C1s (3) [28,29]. For CF-PEI (Figure 4c), three new peaks at approximately 288.7, 287.9 and 285.8 eV could be found, which are attributed to -COOH, -N-C=O and C-N, respectively. The -N-C=O bonds are generated via the amide reaction between -COOH in CF-COOH and the -NH_2_ in PEI molecular [30]. The C–N bonds exist in amino groups of the PEI structure. These results indicate that the branched macromolecule of PEI is chemically grafted rather than coated on the CF surface. The amount of unreacted -NH_2_ in CF-PEI provide a large number of activity points for the next reaction with PDA. For CF-PEI-PDA (Figure 4d), a new peak assigned to -C=N appeared at 287.2 eV, which was produced in Schiff base reaction between the amino groups of PEI and quinone structure of PDA. Besides, N1s high resolution spectrum for CF-PEI-PDA was also fitted in Figure 4g. It could be seen that additional peaks centered at about 399.8 eV, 402.0 eV, 400.0 eV and 398.4 eV, which was corresponding to -C-NH-, -C-NH_2_, -N= and -NH [22,23]. All the XPS results are consistent with the FTIR and Raman results. According to the above analysis, it can be concluded that the “flexible” PEI and “rigid” PDA structure were successfully chemically assembled onto the CF surface.

### 3.3. Surface Wettability of Carbon Fibers

Excellent wetting ability plays an important role in improving the interfacial adhesion between the reinforcement and the matrix. Dynamic contact angle test was conducted to test the contact angles and surface free energy of fibers in polar water and nonpolar diiodomethane, and the results are showed in Figure 5. DCA shows an obvious declining tendency from unmodified CF to CF-PEI and CF-PEI-PDA. Correspondingly, the surface energy increases from 9.9 mN·m^−1^ of unmodified CF to 10.6 mN·m^−1^ of CF-PEI, and dramatically up to 21.0 mN·m^−1^ of CF-PEI-PDA. Surface energy is composed of polar component and dispersion component. As discussed above, grafting PEI and PDA introduced a mass of polar oxygen-containing and nitrogen-containing functional groups, which can greatly increase the polar component. On the other hand, SEM, TEM and AFM results show that the “rigid-flexible” structure composed of PDA and PEI can significantly enhance the surface roughness of fibers, which cause the significant growth of the dispersion component [31]. High surface energy of fibers can improve the wettability and compatibility between CFs and epoxy, thus further enhancing the interfacial adhesion.

### 3.4. Interfacial Properties of CF/EP Composite

The interfacial properties can be assessed by the IFSS and ILSS values of the CF/EP composite, and the results are shown in Figure 6. The IFSS and ILSS values of the unmodified CF/EP composite are only 43.7 and 58.4 MPa because the smooth and chemically inert fibers surface is not conducive to the infiltration by the resin matrix, and the interfacial adhesion between CF and epoxy matrix is weak. After grafting hyperbranched PEI, the IFSS and ILSS values of composites increased to 56.8 and 68.2 MPa, respectively. The main reason is that a stronger interfacial adhesion has been formed due to the improved CF surface wettability, roughness and active reaction points via grafting rich-amino groups hyperbranched PEI. For CF-PEI-PDA/EP, the IFSS and ILSS values reached the maximum of 78.8 MPa and 72.2 MPa, up 80.2% and 23.5% of those of unmodified CF/EP composites, respectively.

The improvement of the interfacial property can be attributed to the following: (1) “Rigid” PDA contained a mass of aromatic structures in its molecular chains, which can further increase the fiber surface roughness and specific surface area, thereby boosting the friction points and contact area between CF-PEI-PDA with epoxy matrix; (2) Plenty of polar functional groups in PDA can greatly enhance the fiber surface energy and the wettability with the matrix, on the one hand. On the other hand, active groups in PDA can not only react with amino groups in PEI but also with the epoxy and curing agent to produce abundant chemical bonds and cross-linked network structure between “rigid” CF and “flexible” matrix. Besides, unreacted amino groups in hyperbranched PEI can also react with the epoxy and curing agent to form physical entanglement with PDA and matrix; (3) The “rigid-flexible” structure composed of PEI and PDA possess an intermediate transition modulus. As an intermediate interface layer, it could effectively avoid the stress concentration due to the huge drop from “rigid” CF to “flexible” epoxy. Wu et al. reported a nacre-like multilayer constructed by alternatively depositing “rigid” polydopamine (PDA) and “flexible” polyether amine (PEA) on carbon fiber surface via the layer-by-layer (LbL) approach. The optimal interfacial strength with three layers of PDA/PEA is 39.2% superior to the unmodified fiber composites [15]. Above IFSS and ILSS results indicate that the double alternant “rigid-flexible” structure have huge advantages and potential for improving the interfacial properties of the composite.

In order to verify the interfacial property of the composites, the failure morphology of CF/EP monofilament was characterized after the micro-drop de-bonding test, and the results are shown in Figure 6. For unmodified CF/EP (Figure 6d), there is almost no residual resin on the surface of carbon fibers and there is almost no residual resin on the fiber surface, indicating that the interfacial adhesion is weak and the failure mode is adhesive failure (Figure 6j). In contrast, some resin particles could be obviously observed on CF-PEI surface, and most grooves become shallow due to the filling of the epoxy (Figure 6e), implying that the interfacial adhesion have been improved and that the failure mode might be the combination of cohesion failure and adhesive failure (Figure 6k). For CF-PEI-PDA/EP, it is obvious that more resins adhere on the fiber surface and the grooves are completely invisible (Figure 6f), which implies that the interfacial adhesion between fibers and matrix have been greatly improved, and that the interfacial failure mode is mainly the cohesion failure (Figure 6l).

The profile section microstructures of the composites, after the interlaminar shear test, are also surveyed to further attest the enhancement of the interfacial properties. As shown in Figure 6g, many interfacial shear slip could be found, a mass of fibers are completely out of the matrix and the exposed fibers are smooth and neat. For CF-PEI/EP (Figure 6h), most fibers are partially embedded into the matrix and partially exposed, with many resin particles sticking on the exposed fiber surface, which indicates that the interfacial interaction between fibers and the matrix becomes stronger, but not strong enough. Comparatively, for the CF-PEI-PDA/EP (Figure 6i), almost all of the fibers are covered with a layer of resin with few gaps between fibers being seen. Besides, there are many large resin chunks protruding on the fiber surface. All these results demonstrate again that the interfacial adhesion of between fibers and the matrix has been greatly improved.

### 3.5. Mechanical Properties of the Composites

The mechanical properties of the composites are evaluated by the impact resistance and flexural performance, and the results are shown in Figure 7a–c. CF-PEI-PDA/EP composites have a maximal impact strength (97.3 kJ·m^−2^), flexural strength (915.5 MPa) and flexural modulus (33.0 GPa), with an improvement of 167.2%, 79.2% and 140.9% comparing with those of the unmodified CF/EP composites, respectively.

The enhancement in the mechanical strength could be verified by the cross-section fracture morphology of the composites. For unmodified CF/EP (Figure 7d), lots of fibers of a certain length were dragged out, with many holes left due to poor interfacial adhesion. After grafting PEI, the amount and length of the fibers pulled out decrease significantly, nonetheless, many holes and cracks could still be seen (Figure 7e). Comparatively, for CF-PEI-PDA/EP, neither fibers pulled out nor holes remained, the fibers and matrix were integrated together tightly, and the cracks between the two disappeared (Figure 7f). All these results imply the enhancement of interface adhesion and mechanical strength of the composites duo to the introduction of the double alternant “rigid-flexible” structure (Figure 7g).

The mechanism for the improvement of the mechanical strength could be illustrated by the following aspects: (1) “Rigid” PDA bridged by “flexible” PEI acting as a barrier layer could deviate the crack propagation path and induce more micro-cracks when the initial cracks passed through, which could consume more energy to break the composite, thus improving the load bearing capacity of the composite. (2) “Flexible” hyperbranched PEI chains enable stiff PDA to cross-lock with each other in multi-dimensional direction, resulting in the plastic deformation, frictional sliding, and more viscoelastic energy dissipation [8]. (3) “Rigid-flexible” structure composed of PDA and PEI endow the interphase region with a moderate modulus between carbon fiber and epoxy resin, which make the external loading uniformly transfer from the matrix to CFs, effectively avoiding the stress concentration due to the huge modulus dropping. (4) Bonds and interactions, such as chemical bonds, hydrogen bonds, π–π interactions and molecular chain entanglement, were formed in the composites via constructing the double alternant “rigid-flexible” structure, which requires plenty of energy to destroy, contributing to the improved mechanical strength.

It is worth noting that the fracture toughness of the composites has been greatly improved from 47.8 J/m^2^ of unmodified CF/EP to 156.4 J/m^2^ of CF-PEI-PDA/EP meanwhile (Figure 6c). The enhancement of fracture toughness could also be inferred by the stress–strain curve of the composites (Figure 7c). For unmodified CF and CF-PEI reinforced composites, the load–displacement curves are linear, which is the characteristic of crisply and low interfacial toughness, while for CF-PEI-PDA, a “kink” point appears in its load–displacement curve, which was caused by the crack initiation. As the load continues increased, the cracks could deflect and propagate by the “rigid” PDA at the interphase, thus enhancing the fracture energy absorption. The typical hackles morphology arising from the shear deformation of matrix on CF-PEI-PDA surface also imply the toughness improvement from the side (Figure 6f).

The mechanism for the improvement of the fracture toughness could be further explained by the Figure 8. Upon the external load, the flexible PEI molecular chains were stretched to line up alongside with the mutual slippage of rigid PDA in the first place. As the load continued to increase, a great deal of hydrogen bonds and covalent bonds between PDA and PEI broke, resulting in abundant energy dissipation together with large strain [32]. With further load increase, the destruction of the π–π conjugated interaction between adjacent PDA also contributes to the energy absorption [33]. All the above analysis indicate that the alternating the dual rigid-flexible structure played an important role in improving the strength and toughness of the composites.

### 3.6. Dynamic Mechanical Thermal Properties of the Composites

The dynamic mechanical thermal properties of the composites were determined by the DMTA technique test, and the results are shown in Figure 9. As shown in Figure 9a, the E′ values of the CF-PEI-PDA/EP composite are higher than those of the unmodified CF/EP composite and CF-PEI/EP over the entire temperature range. The higher E′ value is attributed to the viscoelastic deformation of “flexible” PEI, which results in more energy dissipation to destroy the composites. In addition, the changes in the E′ values demonstrate that the composites reinforced with CF-PEI-PDA could maintain a higher hardness in the high temperature range. As discussed in the previous sections, “rigid” PDA as a shielding layer could deflect a crack propagation path and induce more micro-cracks. What is more, a “rigid” PDA layer could inhibit the molecular chain movement of flexible PEI and epoxy polymers at the interface; consequently, mechanically stiffing the intermediate interface layer, optimizing the thermal resistance property of the composites [34]. Usually, Tg can be determined through the loss factor curve. The temperature at which tanδ shows its maximal value is referred to as Tg [35]. As presented in Figure 9b, Tg increases from 122 °C for the unmodified CF/EP composites to 145 °C for the CF-PEI-PDA/EP composites, indicating that the heat-resistance property of the composites has been improved. In addition, Tanδ is also indicative of the fiber–matrix adhesion at the interface. The lower the tanδ value is, the better the fiber–matrix adhesion. Obviously, the tan δ of CF-PEI-PDA/EP outperforms unmodified CF/EP and CF-PEI/EP, thereby confirming that fiber–matrix adhesion interactions are the strongest of the three by introducing the “rigid and flexible” intermediate structure.

### 3.7. Interface Analysis of the Composites

EDS force modulation equipped on SEM was used to verify the existence of the “rigid-flexible” interface layer with a gradient modulus by assessing the distribution of the carbon element. For unmodified CF (Figure 10a–c), the outline of the CF was clear, and the C element decreases sharply along the yellow arrow, showing an interphase width of about 0.8 μm. After grafting PEI (Figure 10d–f), the outline was clear too, the C element shows the similar downward trends, and the interphase becomes wider with a thickness of 1.0 μm, indicating the presence of a thin PEI layer. Whereas in the cross-section of CF-PEI-PDA (Figure 10g,h,i), the boundary between the CFs and the resin becomes blurred and the interphase thickness increase gradually to 1.6 μm, demonstrating the modulus of the interface exhibits a gradient distribution between the CF and epoxy. The “rigid-flexible” interface with a moderate modulus favors effectively transferring the stress, thereby improving the overall performance of the composite.

In order to verify the advantage of this work, the interfacial strength and fracture toughness improvement of the modified composites in this work, compared with the state-of-the-art in the literature was presented by Figure 11 [15,22,36,37,38,39]. The reported modifications methods included constructing “rigid-flexible” interphase structure via layer-by-layer (LbL) approach using multilayer PDA/polyether amine (CF-(PDA-D400)_3_) (39.2% IFSS increase and 99.8% fracture toughness increase) [15], polyether amine (PEA)/graphene oxide (GO) (CF/(PEA/GO)_9_) (67.7% IFSS increase and 129.0% fracture toughness increase) [37], CF-PDA-PEA with longer chain length of PEA (41.8% IFSS increase) [22]; growing CNT on the CF-PDA by chemical vapor deposition (62.3% IFSS increase) [36], using polydopamine (PDA) as sizing on the surface of carbon fiber (21.0% fracture toughness increase) [38] and PEA/GO (CF-PgP) (94.4% fracture toughness increase) [39]. The effects of all these strategies on the interfacial strength and fracture toughness are inferior to this work (80.2% IFSS increase and 227.2% fracture toughness increase). What is more, constructing “rigid-flexible” structure using PDA and PEI via a simple chemical grafting method for simultaneously greatly strengthening and toughening the interface of carbon fiber/epoxy composite has not been reported yet. That is to say, the design method and modification effect in this work is optimal.

## 4. Conclusions

In this paper, a delicate double alternant “rigid-flexible” structure composed of “rigid” carbon fiber, “flexible” PEI polymer, “rigid” PDA and “flexible” epoxy matrix was elaborated by a simple chemical grafting method. This double alternant “rigid-flexible” structure endows the composites with prominent improvements in both mechanical strength and fracture toughness. Compared with unmodified CF/EP composites, the IFSS value, impact strength and fracture toughness of CF-PEI-PDA/EP composites improved from 43.7 to 78.8 MPa, 36.4 to 97.3 kJ/m^2^ and 47.8 to 156.4 kJ/m^2^, with impressive increases of 80.2%, 167.2% and 227.2%, respectively, superior to the overall properties of other modified composites. The synergistic strengthening and toughening mechanisms of the double alternant “rigid-flexible” structure was discussed in detail. This work provided a promising conception for designing and manufacturing advanced CFRPs structural materials with a desired high strength and toughness.

## Figures and Tables

**Figure 1 nanomaterials-12-03056-f001:**
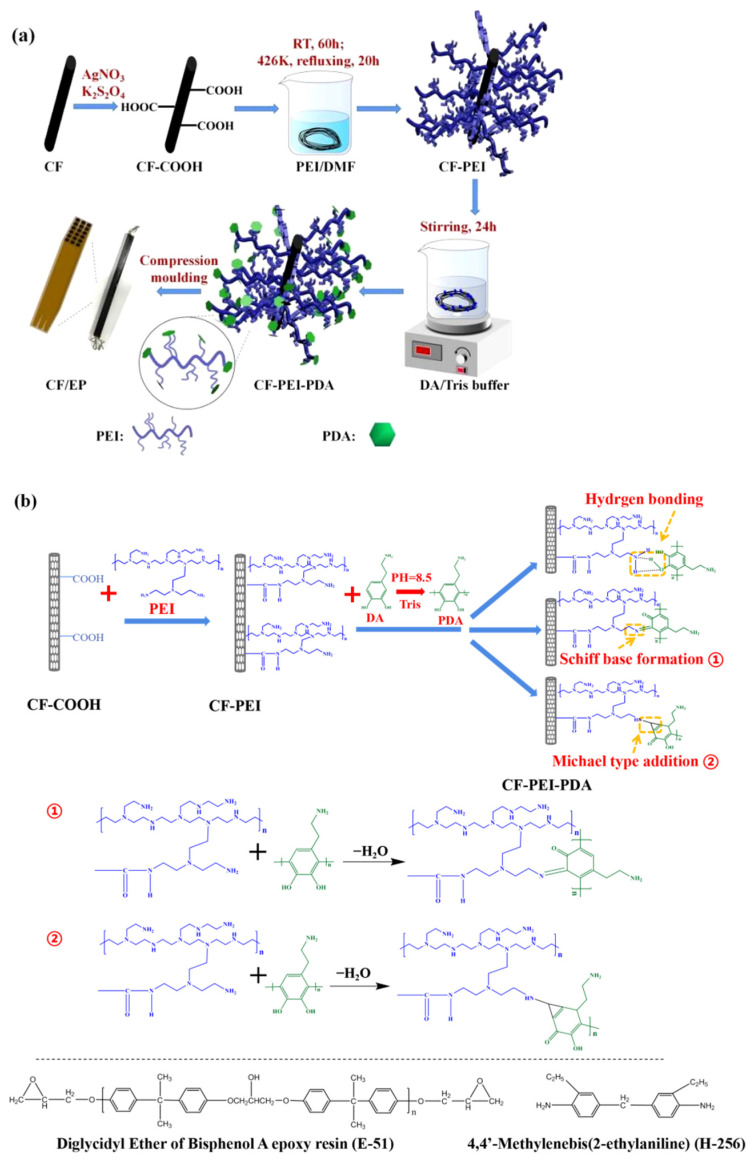
(**a**) Schematic illustration of preparation CF-PEI-PDA and (**b**) main interaction mechanism between CF-PEI and PDA.

**Figure 2 nanomaterials-12-03056-f002:**
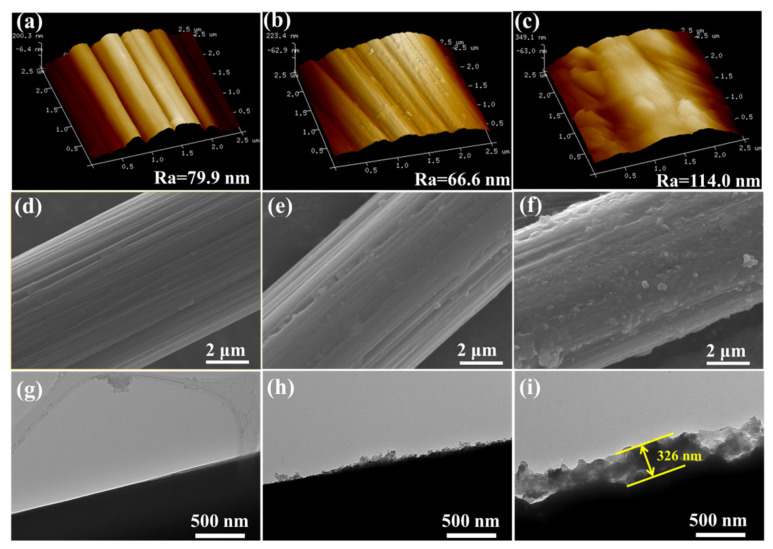
SEM, TEM and AFM images of CF surface topography for: (**a**,**d**,**g**) Unmodified CF; (**b**,**e**,**h**) CF-PEI and (**c**,**f**,**i**) CF-PEI-PDA.

**Figure 3 nanomaterials-12-03056-f003:**
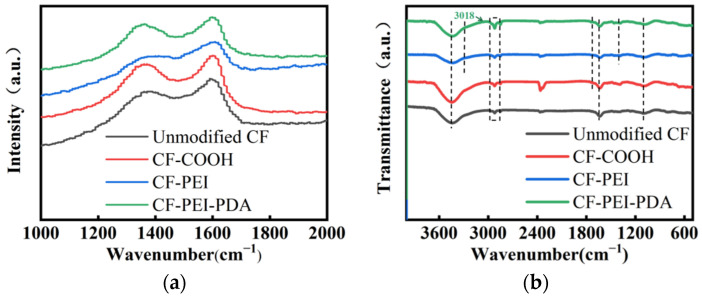
(**a**) Raman spectra for all CFs; the fitting Raman spectra for: (**b**) Unmodified CF.

**Figure 4 nanomaterials-12-03056-f004:**
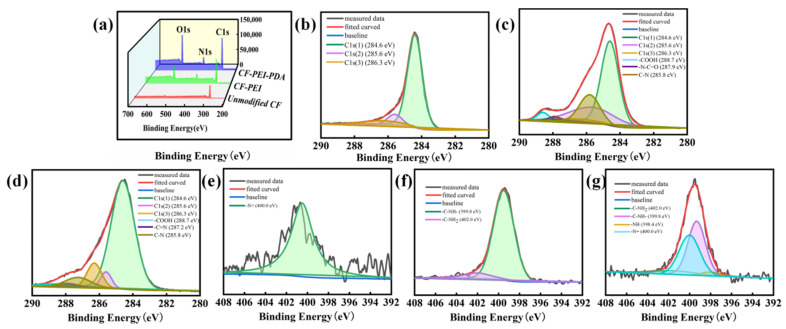
(**a**) Wide-scan XPS spectra for all CFs; C1s and N1s high-resolution XPS spectrum for: (**b**,**e**) Unmodified CF, (**c**,**f**) CF-PEI and (**d**,**g**) CF-PEI-PDA.

**Figure 5 nanomaterials-12-03056-f005:**
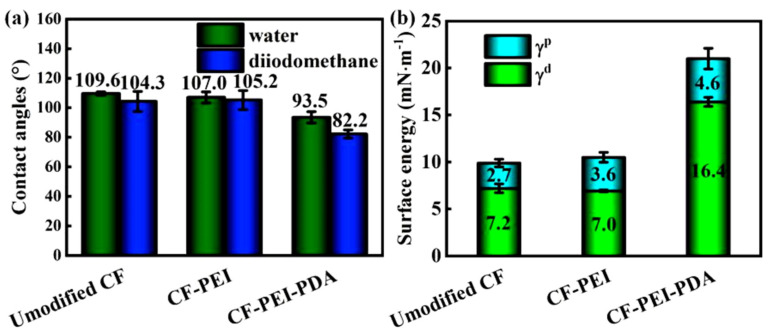
(**a**) Contact angle and (**b**) surface energy of different CFs.

**Figure 6 nanomaterials-12-03056-f006:**
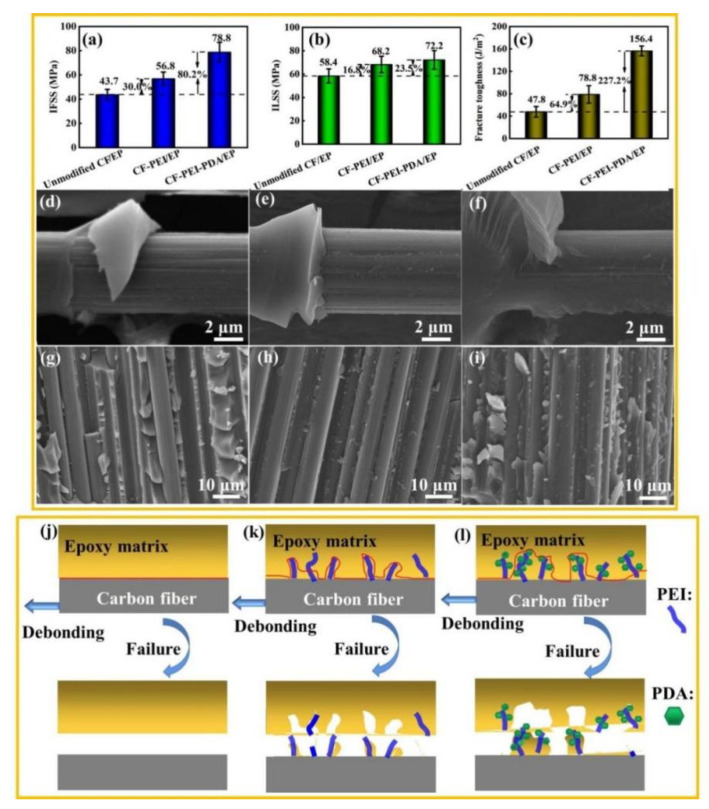
(**a**) IFSS; (**b**) ILSS; (**c**) Fracture toughness; SEM images of de-bonding morphologies of the monofilament, profile-section fracture morphologies of the CF/EP composites and the schematic of failure mode for: (**d**,**g**,**j**) Unmodified CF/EP, (**e**,**h**,**k**) CF-PEI/EP and (**f**,**i**,**l**) CF-PEI-PDA/EP.

**Figure 7 nanomaterials-12-03056-f007:**
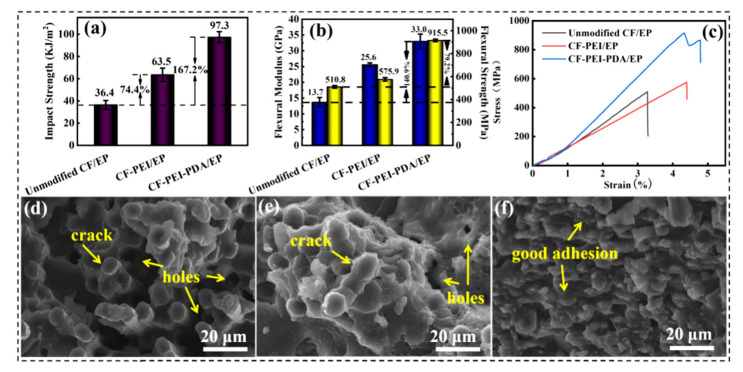
(**a**) Impact strength; (**b**) Flexural strength and flexural modulus; (**c**) Stress–strain curves of different CF/EP composites. SEM images of the fracture morphology for: (**d**) Unmodified CF/EP; (**e**) CF-PEI/EP and (**f**) CF-PEI-PDA/EP; (**g**) Schematic diagram of the strengthening mechanism of CF-PEI-PDA/EP.

**Figure 8 nanomaterials-12-03056-f008:**
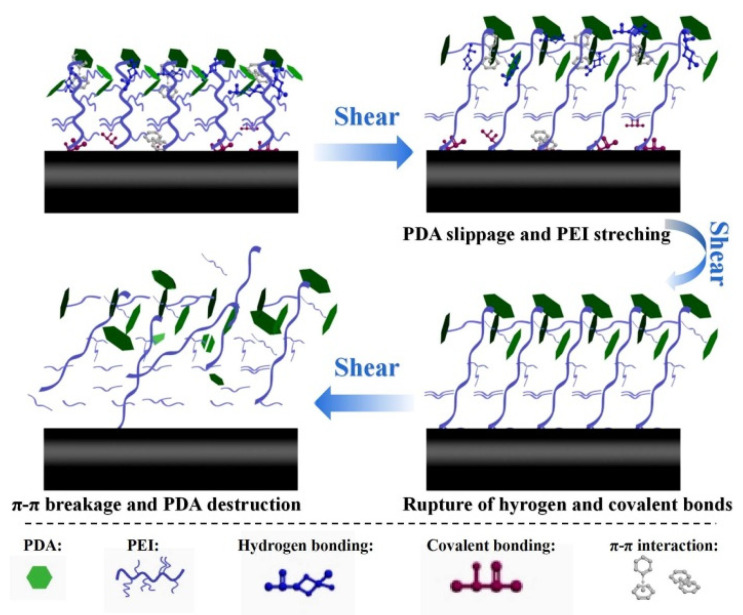
Schematic diagram of toughening mechanism of CF-PEI-PDA/EP composite.

**Figure 9 nanomaterials-12-03056-f009:**
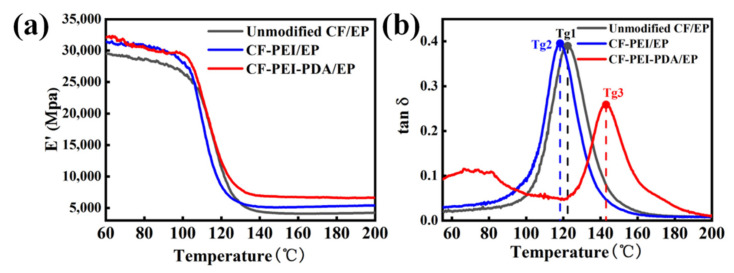
(**a**) Storage modulus (E′) and (**b**) tan δ of different CF/EP composites.

**Figure 10 nanomaterials-12-03056-f010:**
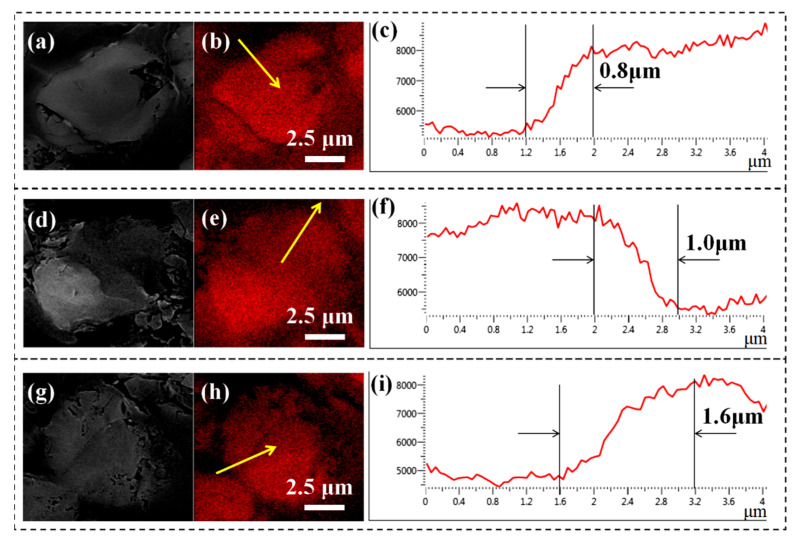
Cross-section SEM images and the carbon element distribution of the CF/EP composite for: (**a**,**b**,**c**) Unmodified CF/EP, (**d**,**e**,**f**) CF-PEI/EP and (**g**,**h**,**i**) CF-PEI-PDA/EP.

**Figure 11 nanomaterials-12-03056-f011:**
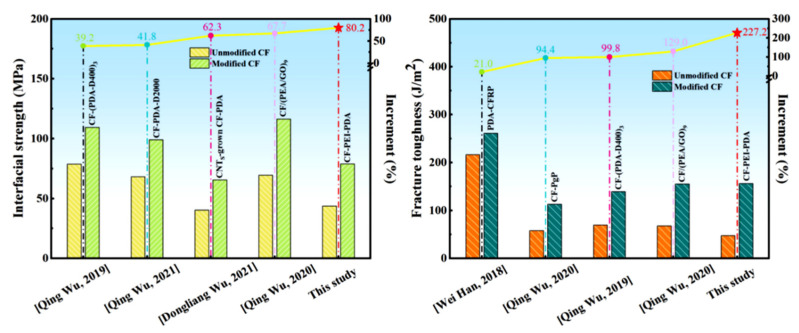
Comparison of interfacial strength and fracture toughness improvement in this work with other composites treated with different sizing agents.

**Table 1 nanomaterials-12-03056-t001:** Raman spectrum information of different carbon fibers.

Samples	D	G	A	I_D_/I_G_	I_A_/I_G_
	W (cm^−1^)	W (cm^−1^)	W (cm^−1^)		
Unmodified CF	1360	1600	1505	1.65	0.21
CF-COOH	1360	1600	1524	1.76	0.24
CF-PEI	1358	1596	1524	1.82	0.25
CF-PEI-PDA	1374	1603	1524	1.93	0.33

**Table 2 nanomaterials-12-03056-t002:** Surface element analysis of different carbon fibers.

Scheme	Element Composition (%)				
	C1s	O1s	N1s	N/C	O/C
Unmodified CF	93.71	4.49	1.80	1.92	4.79
CF-PEI	73.22	17.68	9.10	12.43	24.15
CF-PEI-PDA	67.84	21.79	10.37	15.29	32.12

## Data Availability

The data presented in this study are available on request from the corresponding author.

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
