# Peer review of "Constructing a Double Alternant “Rigid-Flexible” Structure for Simultaneously Strengthening and Toughening the Interface of Carbon Fiber/Epoxy Composites"

_nanomaterials, 2022, doi:10.3390/nano12173056_

Round 1

Reviewer 1 Report

1. Line 16. The authors emphasized the effect of fiber modification on their wettability, therefore, for all described experimental research conditions, it is necessary to specify the humidity of the medium for processes in the air. 

2. Line 100. It is necessary to provide a justification for which criteria and among which other alternatives the parameters diameter of 7 µm density 1.76 g·cm-3 were selected for research.

3. Line 127. It is necessary to specify separately at what exact temperature the mixing, washing and drying of the samples was carried out. 

4. In the article it is necessary for the authors to provide an estimate of the error of the obtained data on the parameters of strength and thermal properties.

Author Response

Response to Reviewer 1 Comments and Suggestions:

Point 1. Line 16. The authors emphasized the effect of fiber modification on their wettability, therefore, for all described experimental research conditions, it is necessary to specify the humidity of the medium for processes in the air.

Response 1. According to your advice, we have supplemented the important information about the humidity in section 2.3.

Point 2. Line 100. It is necessary to provide a justification for which criteria and among which other alternatives the parameters diameter of 7 µm density 1.76 g·cm-3 were selected for research.

Response 2. The fiber selected in the experiment was T300, which was used by many researchers because of its cost-effectiveness.

Point 3. Line 127. It is necessary to specify separately at what exact temperature the mixing, washing and drying of the samples was carried out.

Response 3. The mixing, washing and drying temperatures were 298, 298 and 353 K, respectively, in section 2.2.2.

Point 4. In the article it is necessary for the authors to provide an estimate of the error of the obtained data on the parameters of strength and thermal properties.

Response 4. We have supplemented the information about the number of samples and errors obtained we tested per technique in section 2.3.

Reviewer 2 Report

The article can be published only after serious processing of the chemical part. All chemical processes proposed by the authors should be presented in the form of chemical equations. The reactions shown in Fig. 1 do not fully explain the chemical reactions taking place.

Besides:

-there is no data on what the three buffer (125) is,

- there is no data on the source of dopamine,

-there is no information about the chemical structure of E-51 (107)

- correct the name of DMF on lines 110 and 119

- what does the term "π-orbital hybridization between PEI and PDA (231) mean?

- Raman spectra of different samples do not provide information about changes in the surface structure of carbon fibers and can be included in the accompanying file

- the assignment of samples to unmodified fibers in the text and the experimental part do not coincide (269)

- the absorption band of 2350 cm-1 in the IR spectrum (240) does not belong to carboxyl groups. In addition, no declared carbonyl groups are detected in the spectra

- the formation of Schiff bases and the presence of a Michael reaction (requires vinyl groups) is not proven in any way

- (289) in Table 2, according to photoelectron spectroscopy data, the oxygen content increases sharply after modification of the fibers with polyethylenimine. A very strange phenomenon!

- in order to better understand the results of DMTA, it is necessary to present graphs E' and tg depending on temperature on the same graph. Then it will become clear whether the glass transition temperature of the composite is increasing

- (180) the frequency  DMTA is not specified

Author Response

Response to Reviewer 2 Comments and Suggestions:

The article can be published only after serious processing of the chemical part. All chemical processes proposed by the authors should be presented in the form of chemical equations. The reactions shown in Fig. 1 do not fully explain the chemical reactions taking place.

Response. We have supplemented the chemical equations in details in Fig. 1.

Besides:

Point 1. there is no data on what the three buffer (125) is,

Response 1. The three buffer solution was prepared with 0.01 mol·L-1 Tris and its PH was adjusted to 8.5 with 1 M hydrochloric acid.

Point 2. there is no data on the source of dopamine,

Response 2. More date of dopamine was added in section 2.1 and self-polymerization of dopamine into PDA was in Fig. 1b.

Point 3. there is no information about the chemical structure of E-51 (107)

Response 3. The chemical structure of E-51 was supplied in Fig. 1b.

Point 4. correct the name of DMF on lines 110 and 119

Response 4. We have corrected the name of DMF on lines 110 and 119.

Point 5. what does the term "π-orbital hybridization between PEI and PDA (231) mean?

Response 5. The statement of "π-orbital hybridization between PEI and PDA” was wrong, and we have revised it.

Point 6. Raman spectra of different samples do not provide information about changes in the surface structure of carbon fibers and can be included in the accompanying file

Response 6. According to your advice, we have corrected Fig. 3.

Point 7. the assignment of samples to unmodified fibers in the text and the experimental part do not coincide (269)

Response 7. The unmodified CF in the text and the experimental part are the fibers extracted by acetone.

Point 8. the absorption band of 2350 cm-1 in the IR spectrum (240) does not belong to carboxyl groups. In addition, no declared carbonyl groups are detected in the spectra

Response 8. We have delete the wrong description of “the intensity of the peak at 2350 cm -1 became higher, which was attribute to the increase of C=O in carboxyl groups by oxidation.” The carboxyl groups at 1727 cm-1 could be observed for CF-COOH (240).

Point 9. the formation of Schiff bases and the presence of a Michael reaction (requires vinyl groups) is not proven in any way.

Response 9. We have supplemented the evidences to prove these two reactions in the XPS and FTIR test sections.

Point 10. (289) in Table 2, according to photoelectron spectroscopy data, the oxygen content increases sharply after modification of the fibers with polyethylenimine. A very strange phenomenon!

Response 10. Before grafting PEI, the fibers were treated in the oxidant system of AgNO3/K2S2O4 solution, resulting in the introduction of a large number of oxygen-containing polar groups on CF surface. Reference [23] reported similar results.

Point 11. in order to better understand the results of DMTA, it is necessary to present graphs E' and tg depending on temperature on the same graph. Then it will become clear whether the glass transition temperature of the composite is increasing 

Response 11. When the sample temperature rises to Tg, E’ decreases rapidly, and tan δ has a peak value. In most studies on the CFRPs, Tg was determined through the loss factor depending on temperature curve. As presented in Fig. 9b, Tg increases from 122℃ (Tg1) for the unmodified CF/EP composites to 145℃ (Tg3) for the CF-PEI-PDA/EP composites, and the changing trend of Tg in Fig. 9a was similar.

Point 12. (180) the frequency DMTA is not specified

Response 12. According to your advice, we have presented the load frequency of DMTA (1.0 Hz).

Round 2

Reviewer 2 Report

Manuscript can be accepted in present form